# Novel Structure–Function Models for Estimating Retinal Ganglion Cell Count Using Pattern Electroretinography in Glaucoma Suspects

**DOI:** 10.3390/diagnostics15141756

**Published:** 2025-07-11

**Authors:** Andrew Tirsi, Isabella Tello, Timothy Foster, Rushil Kumbhani, Nicholas Leung, Samuel Potash, Derek Orshan, Celso Tello

**Affiliations:** 1Department of Ophthalmology, Manhattan Eye, Ear, and Throat Hospital, Northwell Health, 210 E 64th St., 8th Floor, New York, NY 10065, USA; isabellatello112@gmail.com (I.T.); ctello@northwell.edu (C.T.); 2Donald and Barbara Zucker School of Medicine at Hofstra/Northwell, Hempstead, NY 11549, USA; tfoster5@northwell.edu (T.F.); rkumbhani@northwell.edu (R.K.); nleung1@northwell.edu (N.L.); 3Albert Einstein College of Medicine, Bronx, NY 10461, USA; samuel.potash@einsteinmed.edu; 4Department of Ophthalmology, Larkin Community Hospital, Miami, FL 33431, USA; derekorshan@gmail.com

**Keywords:** retinal ganglion cells, pattern electroretinogram, glaucoma suspects, optical coherence tomography

## Abstract

**Background/Objectives**: The early detection of retinal ganglion cell (RGC) dysfunction is critical for timely intervention in glaucoma suspects (GSs). The combined structure–function index (CSFI), which uses visual field and optical coherence tomography (OCT) data to estimate RGC counts, may be of limited utility in GSs. This study evaluates whether steady-state pattern electroretinogram (ssPERG)-derived estimates better predict early structural changes in GSs. **Methods**: Fifty eyes from 25 glaucoma suspects underwent ssPERG and spectral-domain OCT. Estimated RGC counts (eRGCC) were calculated using three parameters: ssPERG-Magnitude (eRGCC_Mag_), ssPERG-MagnitudeD (eRGCC_MagD_), and CSFI (eRGCC_CSFI_). Linear regression and multivariable models were used to assess each model’s ability to predict the average retinal nerve fiber layer thickness (AvRNFLT), average ganglion cell layer–inner plexiform layer thickness (AvGCL-IPLT), and rim area. **Results**: eRGCC_Mag_ and eRGCC_MagD_ were significantly correlated with eRGCC_CSFI_. Both PERG-derived models outperformed eRGCC_CSFI_ in predicting AvRNFLT and AvGCL-IPLT, with eRGCC_MagD_ showing the strongest association with AvGCL-IPLT. Conversely, the rim area was best predicted by eRGCC_Mag_ and eRGCC_CSFI_. These findings support a linear relationship between ssPERG parameters and early RGC structural changes, while the logarithmic nature of visual field loss may limit eRGCC_CSFI_’s predictive accuracy in GSs. **Conclusions**: ssPERG-derived estimates, particularly eRGCC_MagD_, better predict early structural changes in GSs than eRGCC_CSFI_. eRGCC_MagD_’s superior performance in predicting GCL-IPLT highlights its potential utility as an early biomarker of glaucomatous damage. ssPERG-based models offer a simpler and more sensitive tool for early glaucoma risk stratification, and may provide a clinical benchmark for tracking recoverable RGC dysfunction and treatment response.

## 1. Introduction

Glaucoma is a neurodegenerative disease of the optic nerve characterized by progressive retinal ganglion cell (RGC) loss, leading to irreversible vision impairment if left untreated [1]. The disease often remains asymptomatic until substantial visual field (VF) damage occurs, which is typically detected using standard automated perimetry (SAP) [1]. However, histological studies have shown that VF defects only become evident once 25% to 50% of RGCs have already been lost [2,3,4,5]. This delay highlights the critical need for early detection strategies that can identify RGC dysfunction before significant irreversible damage occurs.

Glaucoma suspects (GSs) are individuals with clinical features or risk factors for glaucoma, such as elevated intraocular pressure (IOP), suspicious optic nerve head morphology, abnormal retinal nerve fiber layer thickness (RNFLT), or a strong family history of glaucoma [6]. Because GSs often have normal VF results, conventional perimetric testing may not be an effective tool for early disease detection [6]. Optical coherence tomography (OCT) provides structural assessment of RNFLT and ganglion cell layer thickness, and studies suggest that RNFLT loss can be detected approximately six years before VF defects become apparent [7,8,9]. However, structural assessments alone may not reliably detect early RGC dysfunction or preperimetric glaucoma [10,11,12]. Given the limitations of current diagnostic tools and high prevalence of undiagnosed glaucoma, particularly among racial minorities, an objective functional test capable of detecting RGC dysfunction prior to the onset of visual field defects would be invaluable for early intervention [13].

Pattern electroretinogram (PERG) is an electrophysiological test that assesses RGC function by measuring responses to temporally modulated patterned stimuli [14]. Animal studies have demonstrated that PERG signals originate from RGC activity, as optic nerve transection or experimentally induced retinal ischemia abolishes the PERG response while sparing outer retinal function [15,16]. In clinical studies, PERG has shown promise in detecting RGC dysfunction before structural or perimetric abnormalities manifest, with evidence suggesting that PERG abnormalities can precede detectable VF defects by several years [17,18,19]. Specifically, steady-state PERG (ssPERG) has demonstrated superior sensitivity in identifying glaucomatous dysfunction and is easier to implement in clinical settings compared to transient PERG [12,20,21,22].

A key challenge in glaucoma research is the estimation of RGC loss in living patients, as direct histological assessment is only possible post-mortem [21]. To address this limitation, empirical models have been developed to estimate RGC counts (RGCC) using structural and functional parameters. One such model is the combined structure–function index (CSFI), which integrates OCT-derived RNFLT and SAP-derived VF sensitivity to estimate RGCC. CSFI has been shown to closely approximate histological RGCC and outperform individual OCT or SAP measurements in predicting glaucoma progression [3,10]. However, because CSFI relies on perimetric input, its utility in GSs—who have normal VF results—is inherently limited.

Given PERG’s ability to detect RGC dysfunction before structural loss or perimetric defects, an alternative approach to estimating RGCC in GSs may be to develop a PERG-based empirical model. Unlike CSFI, which integrates structural and functional parameters from OCT and SAP, a PERG-based RGCC estimation model would rely on electrophysiological measures of RGC function [12]. This approach has the potential to provide earlier and more accurate identification of individuals at high risk for developing glaucoma, facilitating timely intervention before significant RGC loss occurs.

The purpose of this study was to evaluate the relationships among ssPERG-based estimated RGCC, and SD-OCT structural measures in glaucoma suspects. Specifically, we aimed to compare a PERG-based RGCC estimation model with CSFI and assess their relative abilities to predict glaucomatous damage. Additionally, we sought to determine correlations between PERG-derived RGCC estimates and structural metrics such as RNFLT, ganglion cell layer–inner plexiform layer thickness (GCL-IPLT), and optic nerve head morphology. By exploring these relationships, we hope to establish a novel framework for early glaucoma detection that is independent of perimetric testing and capable of identifying at-risk individuals before irreversible vision loss occurs.

## 2. Materials and Methods

This prospective, cross-sectional study enrolled 25 consecutive glaucoma suspects (50 eyes) from the Manhattan Eye, Ear, and Throat Hospital ophthalmology practice as part of a larger longitudinal study investigating PERG in GSs and glaucoma progression. This study was approved by the institutional review board/Northwell (IRB #18-0397). All subjects signed the informed consent. All participants underwent a comprehensive ophthalmologic examination, which included slit-lamp biomicroscopy, Goldmann applanation tonometry, gonioscopy, SAP using the Humphrey Field Analyzer II (24-2 SITA-Standard strategy) (Carl Zeiss Meditec, Inc., Dublin, CA, USA), spectral-domain optical coherence tomography (SD-OCT) (Carl Zeiss Meditec, Inc., Dublin, CA, USA), and ssPERG (Diopsys Inc., Cedar Knolls, NJ, USA). Participants were between 20 and 80 years old and had a best-corrected visual acuity of 20/40 or better, as measured by Snellen visual acuity testing at the time of enrollment. All subjects were classified as glaucoma suspects based on their initial clinical examination, which included a suspicious glaucomatous optic nerve head appearance defined by an increased cup-to-disc ratio greater than 0.4, neuroretinal rim thinning, or notching or excavation of the optic nerve head. To qualify for inclusion, participants were required to have a repeatable normal Humphrey 24-2 visual field test and no history of intraocular pressure-lowering treatment at the time of enrollment. Individuals with a history of ocular surgery, ocular trauma, or systemic or ocular conditions that could affect optic nerve head structure or function were excluded, except for those with a history of uncomplicated cataract extraction performed less than one year before enrollment.

### 2.1. SD-OCT Testing

The average retinal nerve fiber layer thickness was measured using the Optic Disc Cube protocol of a Cirrus SD-OCT version 6.0 (Cirrus HD-OCT; Carl Zeiss Meditec, Inc., Dublin, CA, USA) [23,24]. The protocol scanned a 6 × 6 mm area centered on the optic nerve head, collecting 200 × 200 axial scans containing 40,000 points. The retinal nerve fiber layer thickness was measured segmentally in quadrants and clock-hour sectors within a 3.46 mm region of interest centered around the optic nerve head and was then averaged. The average ganglion cell-inner plexiform layer thickness was measured using the ganglion cell analysis software program, which, like the Optic Disc Cube protocol, measures ganglion cell-inner plexiform layer thickness segmentally in six wedge-shaped sectors, excluding a 1 mm diameter region around the fovea, and averages the measurements.

### 2.2. ssPERG Testing

Steady-state pattern electroretinography was recorded using the Diopsys NOVAPERG system (Diopsys, Inc., Cedar Knolls, NJ, USA), as previously described [25]. Tests were conducted in a dark room under standardized luminance conditions, free of visual and auditory distractions. Participants’ forehead skin was prepared using NuPrep Skin Prep Gel (Weaver and Company, Aurora, CO, USA), and the lower eyelids were cleaned using OCuSOFT Lid Scrub Original (OCuSOFT Inc., Rosenberg, TX, USA) to ensure optimal conduction. Two disposable hypoallergenic silver/silver chloride skin sensors (Diopsys proprietary skin sensors) were placed on the lower lid margin of both eyes, avoiding the eyelashes, while a third ground electrode (Diopsys EEG electrode) was positioned in the central forehead area with a small amount of conductive paste (Ten20, Weaver and Company). The cables from the Diopsys NOVA device were connected to the electrodes, with each test requiring three electrodes per patient (two active/reference and one ground electrode).

Participants wore the appropriate refractive correction for a viewing distance of 24 inches and were instructed to fixate on a central target displayed on a gamma-corrected Acer V176L bm 17-inch monitor, operating at a refresh rate of 75 frames per second [25]. Luminance output was verified using a MavoSpot 2 USB luminance meter (Gossen GmbH, Nuremberg, Germany). The pattern stimulus consisted of alternating black and white square bars, reversing at 15 reversals per second, with each eye undergoing 25 s trials at high contrast (85%) and low contrast (75%), for a total of 50 s per eye. The stimulus field subtended a visual angle of 1439.90 arc minutes, with each bar subtending 22.49 arc minutes, totaling 64 bars. A red fixation target, subtending 50.79 arc minutes, was centered within the stimulus field. The luminance of the white bars for the 85% and 75% contrast conditions was 204 cd/m^2^, while the luminance of the black bars was 20.5 cd/m^2^ and 52.5 cd/m^2^, respectively, yielding mean luminance values of 112.3 cd/m^2^ and 128.2 cd/m^2^ [25].

All recorded signals underwent band filtration (0.5–100 Hz), amplification (gain = 20,000), and averaging of at least 150 frames. The signal was sampled at 1920 samples per second by an analog-to-digital converter, which had a voltage range programmed between −5 V and +5 V. Sweeps contaminated by eye blinks or gross eye saccades were automatically rejected if they exceeded a threshold voltage of 50 μV, and these segments were marked as artifacts in the report. Synchronized single-channel electroretinography was recorded, generating a time series of 384 data points per analysis frame (200 ms). A fast Fourier transformation was applied to the PERG waveforms to isolate the desired component at 15 reversals per second, with other frequencies, including those originating from eye movements, being filtered out. The PERG test results were stored in an SQL database and presented in report format for analysis. The steady-state PERG protocol used in this study followed the pattern electroretinogram optimized for glaucoma screening (PERGLA) protocol developed by Porciatti et al., which was designed to simplify PERG-assisted glaucoma screening [22]. The PERGLA protocol adhered to the standards set by the International Society for Clinical Electrophysiology of Vision (ISCEV) [26,27,28,29].

For each eye, three PERG parameters were measured: Magnitude, MagnitudeD, and the MagnitudeD/Magnitude ratio. Magnitude (µV) represented the amplitude of the signal at the 15 Hz reversal rate in the frequency domain, while MagnitudeD (µV) represented an adjusted amplitude accounting for phase variability throughout the waveform recording. In eyes with normal retinal ganglion cell function, the response phase remained consistent, resulting in a MagnitudeD value close to that of Magnitude. In eyes with abnormal retinal ganglion cell function, increased response phase variability led to a lower MagnitudeD value relative to Magnitude due to phase cancellation effects. The MagnitudeD/Magnitude ratio provided a within-subject measure of phase consistency. These parameters have been shown to be repeatable, reproducible, and sufficiently reliable for clinical use [26]. Results were presented using a color-coded system similar to a traffic light, with green indicating values within the reference range, yellow representing borderline values, and red denoting results outside the reference range.

### 2.3. Humphrey Field Analyzer Testing

All participants underwent SAP testing using the Humphrey Field Analyzer (HFA) 24-2 protocol. Visual fields tests with more than 20% fixation losses, false-negative and false-positive errors, and mean deviations (MD) < −2 dB were excluded. Using HFA SITA 24-2 results, only participants with visual fields corresponding to stage 0 (no visual field losses) following the Glaucoma Staging System (GSS 2) were enrolled in the study [30].

### 2.4. RGC Count Estimation with the CSFI (eRGCCC_CSFI_)

Estimated RGC count was calculated with the CSFI in accordance with formulas derived by Medeiros et al. [10,31]. The first step involved estimating RGC count using SAP sensitivity values in dB at a given eccentricity, yielding to the first part of the equation, to SAP-derived RGC count (RGCC_SAP_). The slope and intercept of a linear function were used to relate RGC count to sensitivity at a given eccentricity.

The second step involves SD-OCT-derived RGC count (RGCC_OCT_), which was determined with another formula, including AvRNFLT, which was adjusted to age, axonal density (axons/µm^2^), and a correction factor that considered the degree of functional visual impairment to account for RNFL remodeling in advanced disease [10,12,21,31].

The estimated RGC count with the CSFI (eRGCC_CSFI_) model was obtained using the following RGCC_SAP_ and RGCC_OCT_:eRGCCC_CSFI_ = (1 + MD/30) × RGCC_OCT_ + (−MD/30) × SAP_RGCC_

RGC count estimation using PERG parameters:

Magnitude model estimated RGC count (eRGCC_Mag_) and MagnitudeD model for estimated RGC count (eRGCC_MagD_):

Orshan et al. used two generalized linear mixed models to generate eRGCC_Mag_ and eRGCC_MagD_, which can be obtained through the following formulas [12]:eRGCC_Mag_ = 401,342 − (6268 × Age) + (8899 × AvRNFLT) + (58,610 × Mag)eRGCC_MagD_ = 405,529 − (6092 × Age) + (9019 × AvRNFLT) + (53,493 × MagD)

eRGCC_Mag_ and eRGCC_MagD_ were calculated for each study subject by applying the formula.

### 2.5. Statistics

For all variables of interest, outliers with values ≥ 3 standard deviation from the mean were excluded from the analyses. A Shapiro–Wilk test was used to determine normality of the distribution for all important variables. Descriptive statistics were used to evaluate continuous and demographic data. Mean and standard deviations values were determined for all variables of interest. Pearson and partial correlation coefficients were used to test the correlation between eRGCC_CSFI_, eRGCC_Mag_, and eRGCC_MagD_. Independent variables that were significant in exploratory stepwise regression analyses or conceptually important variables based on our review of the literature were included in the final regression analyses. To better understand the relationship between the three eRGCC models with AvGCL-IPLT and AvRNFLT, a linear regression model was used whereby AvGCL-IPLT and then AvRNFLT were entered as dependent variables. After controlling for risk factors for glaucoma, such as age, sex, and central corneal thickness (CCT) (STEP 1), we entered eRGCC_CSFI_, then eRGCC_Mag_, and then eRGCC_MagD_ (STEP 2) as predictors in the regression models. To better assess the relationship between the rim area and the 3 eRGCC models, the rim area was first normalized to the disc area to account for individual variabilities, and the unstandardized revisualized rim area was used in the subsequent analysis, where it was entered as the dependent variable. Age, sex, IOP, and CCT were entered in STEP 1 as covariates, and then the 3 eRGCC models were entered as predictors, one at the time. Statistical analyses were performed with commercially available software (IBM^®^ SPSS^®^ ver. 23.0; SPSS Inc., Chicago, IL, USA).

## 3. Results

### 3.1. Subject Characteristics

Twenty-five glaucoma suspect participants (50 eyes) were recruited and included in the analysis. The characteristics of the study population are summarized in Table 1. The mean age was 54.04 years, and 16 participants were females (64%). The baseline mean HFA MD 24-2 was −0.24 dB and mean IOP was 17.18 mmHg (Table 1).

### 3.2. Consistency Across eRGCC Models and Correlation with OCT Parameters

#### 3.2.1. Correlation Analysis Between eRGCC_CSFI_, eRGCC_Mag_, and eRGCC_MagD_

eRGCC_CSFI_ was significantly correlated with PERG-based eRGCC_Mag_ (Pearson r = 0.950, *p* < 0.001) and with eRGCC_MagD_ (Pearson r = 0.950, *p* < 0.001). This positive correlation was maintained after controlling for age and sex (eRGCC_Mag_: *p* < 0.001, r = 0.879) (eRGCC_MagD_: *p* < 0.001, r = 0.876).

#### 3.2.2. Correlation Analysis Between eRGCC_CSFI_, eRGCC_Mag_, and eRGCC_MagD_, with AvGCL-IPLT and AvRNFLT Measurements

After controlling for age and sex, all three models for eRGCC estimates were significantly correlated with AvRNFLT and AvGCL-IPLT. The correlations were statistically stronger with PERG-based eRGCC, when compared with eRGCC_CSFI_. After residualizing the rim area to the disc size to account for individual variability, eRGCC_CSFI_ correlated better with rim area residuals than PERG-based eRGCC models (Table 2).

### 3.3. Predictive Ability of eRGCC Models

#### 3.3.1. Strength of Association Between eRGCC Models and AvRNFLT

A regression analysis was used to better understand the relationships between eRGCC models and AvRNFLT, where AvRNFLT was entered as the dependent variable, and after controlling for age, sex, and central corneal thickness (Step 1), eRGCC_CSFI_ (step 2) explained 49.6% of the variance in AvRNFLT, with (F (1, 41) = 108.924, *p* < 0.001), whereas in an identical analysis, eRGCCMag and eRGCC_MagD_ explained 63.7% and 63.9% of the variance in AvRNFLT.

(F (1, 41) = 574.474, *p* < 0.001) and (F (1, 41) = 593.742, *p* < 0.001), respectively (Table 3 and Figure 1).

#### 3.3.2. Strength of Association Between eRGCC Models and AvGCL-IPLT

A regression analysis was used to better understand the relationships between eRGCC models and AvGCL-IPLT, where AvGCL-IPLT was entered as the dependent variable, and after controlling for age, sex, and central corneal thickness (Step 1), eRGCC_CSFI_ (step 2) explained 18.9% of the variance in AvGCL-IPLT (F (1, 41) = 16.172, *p* < 0.001), whereas in an identical analysis, eRGCC_Mag_ and eRGCC_MagD_ explained 22.9% and 24.0% of the variance in AvGCL-IPLT, with (F (1, 41) = 21.396, *p* < 0.001), and (F (1, 41) = 22.958, *p* < 0.001), respectively (Table 4 and Figure 2).

#### 3.3.3. Strength of Association Between eRGCC Models and Rim Area

To better understand the relationships between eRGCC models and rim area, a linear regression model was used, where rim area was entered as the dependent variable, and after controlling for age, sex, and CCT (Step 1), eRGCC_CSFI_ (step 2) explained 16.8% of the variance in the rim area (F (1, 21) = 6.732, *p* = 0.017), whereas in an identical analysis, eRGCC_Mag_ and eRGCC_MagD_ explained 15.8% and 13.1% of the variance in the rim area, with (F (1, 21) = 6.230, *p* = 0.021), and (F (1, 21) = 4.908, *p* < 0.038), respectively (Table 5).

## 4. Discussion

In this study, we investigated whether the estimated retinal ganglion cell counts derived from PERG parameters—eRGCC_Mag_ and eRGCC_MagD_—better predicted structural changes in GSs than estimates derived from the combined structure–function index. To our knowledge, this is the first study to directly compare these methods in GSs.

Our findings demonstrated that eRGCC_Mag_ and eRGCC_MagD_ better explained variance in AvRNFLT and AvGCL-IPLT than eRGCC_CSFI_, supporting the hypothesis that ssPERG parameters are better indicators of early structural damage in glaucoma suspects. Notably, eRGCC_MagD_ outperformed eRGCC_Mag_ in predicting GCL-IPL thinning, suggesting that it may better capture early morphological alterations. In contrast, the rim area was best predicted by eRGCC_Mag_ and eRGCC_CSFI_, implying that eRGCC_Mag_ and eRGCC_MagD_ reflect distinct aspects of RGC dysfunction—and that significant axonal loss is required before eRGCC_CSFI_ becomes equally predictive.

CSFI-based eRGCC models have been widely validated [10,32]. Since visual field tests detect glaucoma only after 25–50% of RGCs have been lost, CSFI represents a valuable tool for earlier detection that is superior to SAP and OCT alone [2,3,4,5,10,33]. However, generating eRGCC_CSFI_ requires nine equations and 57 data points per eye [12], and a simpler estimation model would better facilitate clinical integration. Orshan et al. proposed a new formula requiring only three data points per eye to calculate eRGCC_Mag_ and eRGCC_MagD_. Their model performed better than independent PERG and OCT measurements in distinguishing between healthy, GS, and preperimetric glaucoma eyes [12]. Our findings suggest that PERG-derived eRGCC_Mag_ and eRGCC_MagD_ are highly correlated with eRGCC_CSFI_ while also being more predictive of early RGC dysfunction.

PERG-based models do not rely on visual field data, which likely explains their greater accuracy in predicting structural changes in glaucoma suspects, who have no visual defects. The extensive degree of RGC loss prior to identifiable visual defects on SAP may be attributable to the logarithmic scale SAP utilizes to quantify vision loss [34]. Experimental studies have found that utilizing linear units to assess visual acuity resulted in a linear relationship between RGC functional losses and visual field losses, a finding that is further supported by ssPERG studies which have linearly correlated RGC dysfunction with SAP sensitivity losses before logarithmic transformations [3,21,35]. The utilization of SAP’s logarithmic transformation results in an overestimation of RGCC by the CSFI model, as shown in comparisons to histological RGCC [12,21,36,37]. Our results show that excluding SAP data in favor of PERG-based models offers a more accurate prediction of structural changes in the RFNL and GCL-IPL.

The correlation between eRGCC_MagD_ reductions and AvGCL-IPLT thinning aligns with experimental models demonstrating that dendritic atrophy and soma shrinkage precede axonal degeneration [38,39]. This finding builds on prior evidence that functional RGC deficits occur before RGC losses in glaucoma and GSs [19,25,33,40]. In primate and murine models, early dendritic pruning occurs in response to chronic IOP elevation, and prolonged phase latency—captured by eRGCC_MagD_ —has been linked to these morphological changes [41]. Given that GCL-IPL contains RGC somas and dendrites, while the RNFL consists of axons, our results reinforce the concept that GCL-IPL thinning is an early event in glaucoma pathogenesis, detectable before measurable RNFL loss [21].

Tirsi et al. demonstrated PERG’s ability to detect early RGC changes before cell death [20,21,42,43]. Specifically, increased phase latency in dysfunctional GS eyes has been linked to dendritic pruning, which precedes RNFL thinning. Since RGC somas and dendrites reside in the GCL-IPL, while axons are in the RNFL, these findings support the hypothesis that GCL-IPL thinning is an earlier marker of glaucomatous damage in GSs [21].

A study by Tirsi et al. (2022) found a significant correlation between eRGCC_CSFI_ and PERG parameters (Mag and MagD) [21]. A mediation analysis suggested that eRGCC_CSFI_ mediated the relationship between PERG and structural measures like AvRNFLT and AvGCL-IPLT, supporting the idea that dysfunctional RGCs exhibit reduced electrical response amplitudes and prolonged phase latency, which correspond with soma shrinkage, dendritic thinning, and axonal diameter reduction [21]. These findings suggest Mag and MagD assess distinct aspects of RGC function: Mag reflects the overall electrical response, while MagD represents phase latency, an indicator of soma, axon, and dendritic integrity, and highlight the importance of RGC structural integrity for both cell viability and electrophysiological function.

Our study demonstrates that eRGCC_Mag_ and eRGCC_MagD_ are stronger predictors of AvRNFLT and AvGCL-IPLT than eRGCC_CSFI_. In contrast, eRGCC_CSFI_ explained more variance in the rim area compared to eRGCC_Mag_ and eRGCC_MagD_. Since the SAP that eRGCC_CSFI_ incorporates was noncontributory in our study population, and the amount of variance in the rim area predicted by all modalities was much lower compared to AvRNFLT and AvGCL-IPLT, these results likely reflect a decreased ability of each model to predict this specific glaucomatous structural change. However, findings from Patel et al. may provide some explanation for this contrasting result. This study demonstrated that the relationship between RNFLT and the minimum neural rim width is nonlinear and logarithmic, similarly to the logarithmic scale used by SAP to quantify visual loss. This relationship was consistent even after adjusting for the unit of measurement and was true for the rim volume as well [44]. Previous studies have shown that PERG-based models have a linear relationship with visual loss [3,21,35]. Together, this suggests that AvRNFLT and AvGCL-IPLT have a linear relationship with the progression of glaucomatous disease, while structural changes to the neural rim have a logarithmic one. This could explain why eRGCC_Mag_ and eRGCC_MagD_, which progress linearly, struggle to explain variance in the rim area in GSs. It also suggests that eRGCC_CSFI_ is a potentially robust predictor of the rim area in diagnosed glaucoma patients with visual field defects, given SAP and changes to the neural rim both appear to progress along a logarithmic scale.

We report that the average eRGCC_Mag_ for our subjects was approximately 970,759, while the average eRGCC_MagD_ was 971,665, indicating a difference of 906 cells. We hypothesize that this difference represents the number of dysfunctional RGCs that are still alive but not functioning properly. This is based on the understanding that eRGCC_Mag_ estimates the number of healthy, functioning RGCs, whereas eRGCC_MagD_ includes both healthy and dysfunctional cells [21]. Experimental and clinical evidence suggests that RGCs, particularly in the early stages of glaucoma, may retain the capacity to recover function after periods of dysfunction [45]. Therefore, this difference may serve as a clinically meaningful benchmark for quantifying the number of “sick” RGCs. For example, comparing the difference between eRGCC_MagD_ and eRGCC_Mag_ before and after treatment may help determine whether the number of dysfunctional RGCs has decreased, offering a potential marker for therapeutic response.

### Limitations

In this study the cohort was relatively small, consisting of 25 subjects (50 eyes), and larger studies are needed to confirm these findings. Longitudinal data points were not analyzed in this study. Additionally, our analysis used GCL-IPL thickness measurements, which combine ganglion cell layer soma and inner plexiform layer dendritic arbor measurements into a single value. Future investigations should consider utilizing other OCT devices capable of separately quantifying these structures. Further research is also needed to validate the new PERG-derived eRGCC models across different populations and disease severities.

## 5. Conclusions

In this study, eRGCC_Mag_ and eRGCC_MagD_ were significantly correlated with eRGCC_CSFI_, even after adjusting for age and sex. When predicting variance in AvRNFLT, eRGCC_Mag_ and eRGCC_MagD_ outperformed eRGCC_CSFI_. Similarly, eRGCC_Mag_ and eRGCC_MagD_ better explained the variability in AvGCL-IPLT, with eRGCC_MagD_ performing slightly better than eRGCC_Mag_. These findings align with previous observations that RGC soma and dendritic changes precede axonal damage, reinforcing the hypothesis that eRGCC_MagD_, which reflects these morphological changes, should better predict AvGCL-IPLT thinning.

When evaluating the rim area, eRGCC_Mag_ and eRGCC_CSFI_ exhibited comparable predictive power, suggesting that axonal damage plays a greater role in rim area changes. These findings further support the concept that Mag and MagD measure distinct aspects of RGC function, with Mag representing RGC electrical response amplitude and MagD reflecting the temporal dynamics of RGC neural pathways. Taken together, eRGCC_Mag_ and eRGCC_MagD_ provide complementary models for assessing RGC function in early glaucomatous disease, while the findings relating to rim area suggest that more severe damage is required before eRGCC_CSFI_ becomes equally predictive.

## Figures and Tables

**Figure 1 diagnostics-15-01756-f001:**
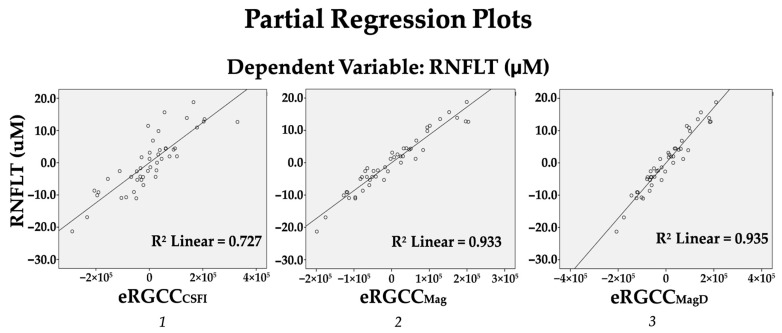
Linear regression models illustrating the relationship between estimated retinal ganglion cell count (eRGCC) and average retinal nerve fiber layer (RNFL) thickness after adjusting for age, sex, and central corneal thickness. (***1***) eRGCC_CSFI_ shows a strong positive correlation with RNFL thickness. (***2***) eRGCC_Mag_ derived from steady-state PERG amplitude, and (***3***) eRGCC_MagD_ derived from PERG amplitude and phase, with each showing an even stronger correlation with RNFL thickness, suggesting improved sensitivity to structural damage.

**Figure 2 diagnostics-15-01756-f002:**
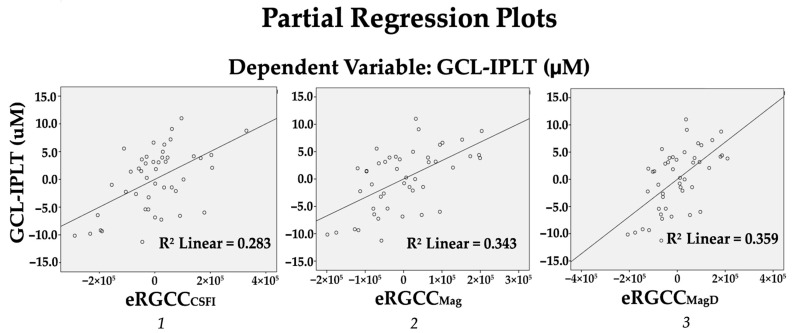
Linear regression models illustrating the relationship between estimated retinal ganglion cell count (eRGCC) and average ganglion cell layer–inner plexiform layer (GCL-IPL) thickness after adjusting for age, sex, and central corneal thickness. (***1***) eRGCC_CSFI_ shows a moderate positive correlation with GCL-IPL thickness. (***2***) eRGCC_Mag_ derived from steady-state PERG amplitude, and (***3***) eRGCC_MagD_ derived from PERG amplitude and phase, with each showing a stronger correlation with GCL-IPL thickness, suggesting improved sensitivity to structural damage.

**Table 1 diagnostics-15-01756-t001:** Demographic and clinical characteristics of participants.

*N* = 50 Eyes (25 Subjects)	Mean ± Standard Deviation
Age (years)	54.04 ± 15.69
Sex (% females)	9 M/16 F (64%)
IOP (mmHg)	17.18 ± 3.71
24-2 MD (dB)	−0.24 ± 1.26
Central corneal thickness (µm)	546.43 ± 29.67
Spherical equivalent (diopter)	−0.93 ± 2.64
Disc area (mm^2^)	1.94 ± 0.42
Rim area (mm^2^)	1.13 ± 1.63
eRGCC_CSFI_	969,050.14 ± 189,230.41
eRGCC_Mag_	970,759.34 ± 181,022.09
eRGCC_MagD_	971,665.90 ± 180,955.78

IOP—intraocular pressure; MD—mean deviation; eRGCC_CSFI_—estimated retinal ganglion cell count based on the combined structure–function index; eRGCC_Mag_—estimated retinal ganglion cell count based on Magnitude; eRGCC_MagD_—estimated retinal ganglion cell count based on MagnitudeD.

**Table 2 diagnostics-15-01756-t002:** Partial correlation analysis between three estimated retinal ganglion cells counts models and different thickness and rim area measurements, controlling for age and sex.

	AvRNFLT	AvGCL-IPLT	Rim Area (mm^2^)	Rim Area (Residuals)
eRGCC_CSFI_	0.844 **	0.553 **	0.342	0.426 *
eRGCC_Mag_	0.956 **	0.560 **	0.267	0.365 °
eRGCC_MagD_	0.957 **	0.575 **	0.247	0.348 °

AvRNFLT—average retinal nerve fiber layer thickness; AvGCL-IPLT—average ganglion cell-inner plexiform layer thickness; eRGCC_CSFI_—estimated retinal ganglion cell count based on the combined structure–function index; eRGCC_Mag_—estimated retinal ganglion cell count based on Magnitude; eRGCC_MagD_—estimated retinal ganglion cell count based on MagnitudeD. ** *p* < 0.001, * *p* < 0.05, ° 0.05 < *p* < 0.99.

**Table 3 diagnostics-15-01756-t003:** Associations of eRGCC_CSFI_, eRGCC_Mag_, and eRGCC_MagD_ with AvRNFLT, controlling for age, sex, and central corneal thickness.

	Step 1 (age, sex, and CCT)		Step 2 (eRGCC_CSFI_)			
	ΔR^2^	B (95% CI)	ΔR^2^	B (95% CI)	R^2^	SE
AvRNFLT	0.317 *	124.61 (70.82, 178.41)	0.496 **	6.228 × 10^−5^ (0.00, 0.00)	0.813	4.82
	Step 1 (age, sex, and CCT)		Step 2 (eRGCC_Mag_)			
	ΔR^2^	B (95% CI)	ΔR^2^	B (95% CI)	R^2^	S.E.
AvRNFLT	0.317 *	124.61 (70.82, 178.41)	0.637 **	8.628 × 10^−5^ (0.00, 0.00)	0.955	2.38
	Step 1 (age, sex, and CCT)		Step 2 (eRGCC_MagD_)			
	ΔR^2^	B (95% CI)	ΔR^2^	B (95% CI)	R^2^	S.E.
AvRNFLT	0.317 *	124.61 (70.82, 178.41)	0.639 **	8.565 × 10^−5^ (0.00, 0.00)	0.956	2.34

Steps of the regression are shown separated by the columns. ΔR^2^ is the change in R^2^; B (95% CI) is the B coefficient and 95% confidence interval ranges; R^2^ is the total R^2^ of the model; S.E. is the standard error of the estimate of the final model. CCT—central corneal thickness; AvRNFLT—average retinal nerve fiber layer thickness; eRGCC_CSFI_—estimated retinal ganglion cell count based on the combined structure–function index; eRGCC_Mag_—estimated retinal ganglion cell count based on PERG Magnitude; eRGCC_MagD_—estimated retinal ganglion cell count based on PERG MagnitudeD. ** *p* < 0.01, * *p* < 0.05.

**Table 4 diagnostics-15-01756-t004:** Associations of eRGCC_CSFI_, eRGCC_Mag_, and eRGCC_MagD_ with GCL-IPLT, controlling for age, sex, and central corneal thickness.

	Step 1 (age, sex, and CCT)		Step 2 (eRGCC_CSFI_)			
	ΔR^2^	B (95% CI)	ΔR^2^	B (95% CI)	R^2^	SE
AvGCL_IPLT	0.331 *	129.45 (94.92, 164.09)	0.189 **	2.502 × 10^−5^ (0.00, 0.00)	0.521	5.03
	Step 1 (age, sex, and CCT)		Step 2 (eRGCC_Mag_)			
	ΔR^2^	B (95% CI)	ΔR^2^	B (95% CI)	R^2^	SE
AvGCL_IPLT	0.331 *	129.45 (94.82, 164.09)	0.229 **	3.367 × 10^−5^ (0.00, 0.00)	0.561	4.81
	Step 1 (age, sex, and CCT)		Step 2 (eRGCC_MagD_)			
	ΔR^2^	B (95% CI)	ΔR^2^	B (95% CI)	R^2^	SE
AvGCL_IPLT	0.331 *	129.45 (94.82, 164.09)	0.240 **	3.416 × 10^−5^ (0.00, 0.00)	0.571	4.75

Steps of the regression are shown separated by the columns. ΔR^2^ is the change in R^2^; B (95% CI) is the B coefficient and 95% confidence interval ranges; R^2^ is the total R^2^ of the model; SE is the standard error of the estimate of the final model. CCT—central corneal thickness; AvGCL-IPLT—average ganglion cell layer–inner plexiform layer thickness; eRGCC_CSFI_—estimated retinal ganglion cell count based on the combined structure–function index; eRGCC_Mag_—estimated retinal ganglion cell count based on PERG Magnitude; eRGCC_MagD_—estimated retinal ganglion cell count based on PERG MagnitudeD. ** *p* < 0.01, * *p* < 0.05.

**Table 5 diagnostics-15-01756-t005:** Associations of eRGCCCSFI, eRGCCMag, and eRGCCMagD with rim area, controlling for age, sex, and CCT.

	Step 1 (age, sex, and CCT)		Step 2 (eRGCC_CSFI_)			
	ΔR^2^	B (95% CI)	ΔR^2^	B (95% CI)	R^2^	SE
Rim area	0.309 *	0.136 (−0.118, 0.391)	0.168 *	6.365 × 10^−7^ (0.00, 0.00)	0.477	1.26
	Step 1 (age, sex, and CCT)		Step 2 (eRGCC_Mag_)			
	ΔR^2^	B (95% CI)	ΔR^2^	B (95% CI)	R^2^	SE
Rim area	0.309 *	0.136 (−0.118, 0.391)	0.158 *	7.430 × 10^−7^ (0.00, 0.00)	0.467	0.13
	Step 1 (age, sex, and CCT)		Step 2 (eRGCC_MagD_)			
	ΔR^2^	B (95% CI)	ΔR^2^	B (95% CI)	R^2^	SE
Rim area	0.309 *	0.136 (−0.118, 0.391)	0.131 *	6.623 × 10^−7^ (0.00, 0.00)	0.440	1.31

Steps of the regression are shown separated by the columns. ΔR^2^ is the change in R^2^; B (95% CI) is the B coefficient and 95% confidence interval ranges; R^2^ is the total R^2^ of the model; SE is the standard error of the estimate of the final model. CCT—central corneal thickness; AvGCL-IPLT—average ganglion cell layer–inner plexiform layer thickness; eRGCC_CSFI_—estimated retinal ganglion cell count based on the combined structure–function Index; eRGCC_Mag_—estimated retinal ganglion cell count based on PERG Magnitude; eRGCC_MagD_—estimated retinal ganglion cell count based on PERG MagnitudeD. * *p* < 0.05.

## Data Availability

The data presented in this study are available on request from the corresponding author due to patient privacy.

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
