# Peer review of "Novel Structure–Function Models for Estimating Retinal Ganglion Cell Count Using Pattern Electroretinography in Glaucoma Suspects"

_diagnostics, 2025, doi:10.3390/diagnostics15141756_

Round 1

Reviewer 1 Report

Comments and Suggestions for Authors

This study is aimed to evaluate the relationships among ssPERG-based estimated RGCC, and SD-OCT structural measures in glaucoma suspects, and to compare a PERG-based RGCC estimation model with CSFI and assess their relative abilities to predict glaucomatous damage. This is the first study to directly compare the estimated retinal ganglion cell counts derived from PERG parameters--eRGCCMag and eRGCCMagD than estimates derived from the combined structure-function index in glaucoma suspect (GS). Authors reported that PERG-derived eRGCCMag and eRGCCMagD are highly correlated with eRGCCCSFI while also being more predictive of early RGC dysfunction in GS. And their results  reinforce the concept that GCL-IPL thinning is an early event in glaucoma pathogenesis, detectable before measurable RNFL loss and the disc rim decreases.

Overall, this study is well-designed and interpretation, and the  eRGCCMag and eRGCCMagD were significantly correlated with eRGCCCSFI, which offers a diagnosis aid in detecting GS. 

The only shortage is limited small enrolled number, and no longitudinal data to confirm this estimation. 

Second, Line233-235 should be deleted in results section

Author Response

Full Reviewer 1 Comment: 

This study is aimed to evaluate the relationships among ssPERG-based estimated RGCC, and SD-OCT structural measures in glaucoma suspects, and to compare a PERG-based RGCC estimation model with CSFI and assess their relative abilities to predict glaucomatous damage. This is the first study to directly compare the estimated retinal ganglion cell counts derived from PERG parameters--eRGCCMag and eRGCCMagD than estimates derived from the combined structure-function index in glaucoma suspect (GS). Authors reported that PERG-derived eRGCCMag and eRGCCMagD are highly correlated with eRGCCCSFI while also being more predictive of early RGC dysfunction in GS. And their results  reinforce the concept that GCL-IPL thinning is an early event in glaucoma pathogenesis, detectable before measurable RNFL loss and the disc rim decreases.

Overall, this study is well-designed and interpretation, and the  eRGCCMag and eRGCCMagD were significantly correlated with eRGCCCSFI, which offers a diagnosis aid in detecting GS. 

The only shortage is limited small enrolled number, and no longitudinal data to confirm this estimation. 

Second, Line233-235 should be deleted in results section

Comment 1: [The only shortage is limited small enrolled number, and no longitudinal data to confirm this estimation. ]
Response 1: Thank you for pointing this out. We agree with this comment. Therefore, we have revised our limitations section to address the lack of longitudinal data in this analysis.

This change can be found on page 14, paragraph 2, and line 431.]"[Limitations In this study the cohort was relatively small, consisting of 25 subjects (50 eyes), and larger studies are needed to confirm these findings. Longitudinal data points were not analyzed in this study. Addition- ally, our analysis used GCL-IPL thickness measurements, which combine ganglion cell layer soma and inner plexiform layer dendritic arbor meas- urements into a single value. Future investigations should consider utiliz- ing other OCT devices capable of separately quantifying these structures. Further research is also needed to validate the new PERG-derived eRGCC models across different populations and disease severities.]"

Comments 2: [Second, Line233-235 should be deleted in results section]

Response 2: Agree. We have, accordingly, removed this text, which was included from the journal template in error.

We thank reviewer 1 for their time and effort in critically reading our manuscript and are grateful for their efforts in helping us to improve it. 

Reviewer 2 Report

Comments and Suggestions for Authors

In patients with glaucoma suspect an objective functional test capable of detecting RGC dysfunction prior to the onset of visual field defects would be invaluable for early intervention. Pattern Electroretinogram (specifically, steady-state PERG (ssPERG, 3 parameters)) can precede detectable VF defects by several years. The purpose of this prospective, cross-sectional study (50 eyes of 25 subjects with glaucoma suspect) was to evaluate the relationships among ssPERG-based estimated retinal ganglion cell count (eRGCC), and parameters of SD-OCT structural measures, including/versus the combined structure-function index (CSFI), in glaucoma suspects. The findings demonstrated that ssPERG parameters were more indicators (with high correlation to CSFI) of early structural damage in glaucoma suspects, not relying on visual field data. These results are of clinical value and the manuscript is very interesting to read. Great performance.

Material and Methods: I am sure gonioscopy was performed as well.

Author Response

Comments 1: [In patients with glaucoma suspect an objective functional test capable of detecting RGC dysfunction prior to the onset of visual field defects would be invaluable for early intervention. Pattern Electroretinogram (specifically, steady-state PERG (ssPERG, 3 parameters)) can precede detectable VF defects by several years. The purpose of this prospective, cross-sectional study (50 eyes of 25 subjects with glaucoma suspect) was to evaluate the relationships among ssPERG-based estimated retinal ganglion cell count (eRGCC), and parameters of SD-OCT structural measures, including/versus the combined structure-function index (CSFI), in glaucoma suspects. The findings demonstrated that ssPERG parameters were more indicators (with high correlation to CSFI) of early structural damage in glaucoma suspects, not relying on visual field data. These results are of clinical value and the manuscript is very interesting to read. Great performance.

Material and Methods: I am sure gonioscopy was performed as well.]

Response 1: Thank you for pointing this out. We agree with this comment.
Therefore, we have included in our material methods section the following on [Explain what change you have made. Mention exactly where in the revised manuscript this change can be found -
page 4, paragraph 1, and line 102.]" [All participants underwent a comprehensive ophthalmologic examination, which included slit-lamp biomicroscopy, Goldmann applanation tonometry, gonioscopy, SAP using the Humphrey Field Analyzer II (24-2 SITA-Standard strategy), spectral-domain optical coherence tomography (SD-OCT) (Carl Zeiss Meditec, Inc., Dub- lin, CA, USA), and ssPERG (Diopsys Inc., Cedar Knolls, NJ, USA). ]"

We thank reviewer 2 for their time and effort in critically reviewing our manuscript and for helping us to improve it.